# Extracting electronic many-body correlations from local measurements with artificial neural networks

Faluke Aikebaier[1,2,3], Teemu Ojanen[1,2] and Jose L. Lado[3]

**1** Computational Physics Laboratory, Physics Unit, Faculty of Engineering and Natural Sciences, Tampere University, FI-33014 Tampere, Finland
**2** Helsinki Institute of Physics P.O. Box 64, FI-00014, Finland
**3** Department of Applied Physics, Aalto University, 00076, Espoo, Finland

## Abstract

The characterization of many-body correlations provides a powerful tool for analyzing correlated quantum materials. However, experimental extraction of quantum entanglement in correlated electronic systems remains an open problem in practice. In particular, the correlation entropy quantifies the strength of quantum correlations in interacting electronic systems, yet it requires measuring all the single-particle correlators of a macroscopic sample. To circumvent this bottleneck, we introduce a strategy to obtain the correlation entropy of electronic systems solely from a set of local measurements. We show that, by combining local particle-particle and density-density correlations with a neural-network algorithm, the correlation entropy can be predicted accurately. Specifically, we show that for a generalized interacting fermionic model, our algorithm yields an accurate prediction of the correlation entropy from a set of noisy local correlators. Our work shows that the correlation entropy in interacting electron systems can be reconstructed from local measurements, providing a starting point to experimentally extract many-body correlations with local probes.

# 1 Introduction

Quantum correlations between interacting particles is one of the most fundamental aspects of many-body theory [1]. The strength of quantum correlations quantifies the complexity of a many-body state with respect to a product state of non-interacting particles [2–6]. Specifically, non-interacting fermionic systems are described by Slater determinant states with vanishing multi-particle entanglement. While highly correlated states emerge from many-body electronic interactions, strong interactions may or may not guarantee strong quantum many-body correlations. In particular, while some quantum states are faithfully captured by a mean-field theory [7,8], more exotic correlated states inherently emerge from non-trivial quantum correlations [9,10]. In this spirit, many-body entanglement provides a powerful framework to classify emergent properties of quantum materials [11,12]. However, direct experimental probes of entanglement remain very limited [13].

Quantum correlations in a many-body state can be quantified by different entanglement measures, including the bipartite von Neumann and Renyi entropies of the density matrix [14–16]. These quantities characterize quantum correlations between two arbitrarily chosen disjoint partitions of a system, yet importantly, are non-zero even for non-interacting electronic systems [17–20]. Instead, to quantify entanglement in fermionic systems, it is natural to adopt the von Neumann entropy associated with the correlation matrix, or one-particle density matrix (ODM) [21–24], that identically vanishes in the non-interacting limit. The correlation entropy has been theoretically studied in various fermionic systems [25–30], however, it has not been experimentally measured up to date. Such measurement would require extracting two-point correlations in the whole system, which is unfeasible for thermodynamically large systems.

Here we address the gap between theory and experiments in electronic many-body entanglement by designing a methodology to obtain the correlation entropy from minimal local measurement data. This approach is illustrated in Fig. 1(a). The input data for the algorithm constitutes particle-particle and density-density correlation functions on a set of sites. The neural-network algorithm allows extracting the correlation entropy from the set of local measurements. Ultimately, if obtained experimentally, this would allow extraction of the correlation entropy of the physical system We demonstrate this methodology by analyzing a generalized interacting model incorporating local and non-local interactions, spin-orbit coupling, and magnetic fields. We show that our procedure is robust to noise in the input data, providing a promising strategy to characterize the strength of many-body correlations in interacting electronic systems.

# 2 Model

Before addressing our approach based on local measurements, we first illustrate how the correlation entropy can be computed theoretically [Fig. 1(b)], where we access to all the non-local correlation functions. In the following, we take the correlation entropy as the characteristic measure of the strength of many-body entanglement in fermionic systems. The correlation entropy is a type of many-body fermionic entanglement entropy similar to partition entanglement entropy

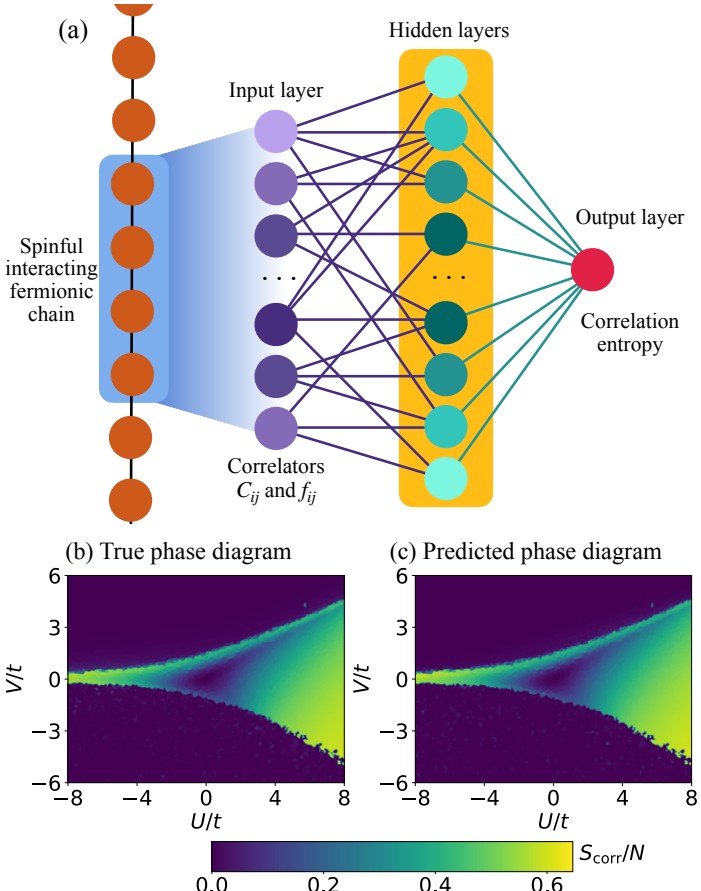

Figure 1: The neural-network structure and the phase diagram of the correlation entropy. (a) Schematic of a one-dimensional interacting fermionic system, where a set of local measurements allow extracting the correlation entropy through a trained neural-network algorithm. (b) The $UV$-phase diagram of the correlation entropy. (c) The neural-network model predicted $UV$-phase diagram of the correlation entropy.

in spin systems. The correlation entropy in a fermionic many-body state $|\Psi_0\rangle$ is defined in terms of the correlation matrix

$$C_{ij}^{ss'} = \langle\Psi_0| c_{is}^{\dagger} c_{js'} |\Psi_0\rangle, \tag{1}$$

where $c_{is}$, $c_{is}^{\dagger}$ are the annihilation and creation operators at site $i$ and spin $s$. For a system with $N$ spinful sites, the correlation matrix has dimensions $2N \times 2N$. Due to the Fermi statistics, the eigenvalues $\alpha_i$ of the correlation matrix satisfy $0 \le \alpha_i \le 1$. The definition in Eq. (1) is similar to the Green's function matrix, but the convenience of the correlation matrix allows to define the correlation entropy as

$$S_{\text{corr}} = -\sum_{j=1}^{2N} \alpha_j \log \alpha_j, \tag{2}$$

which is non-negative and, in contrast to partition entanglement entropies [14], scales linearly in the system size even in ground states of local Hamiltonians. The correlation entropy is a measure of fermionic entanglement in the following sense. If there exists a single-particle basis in which

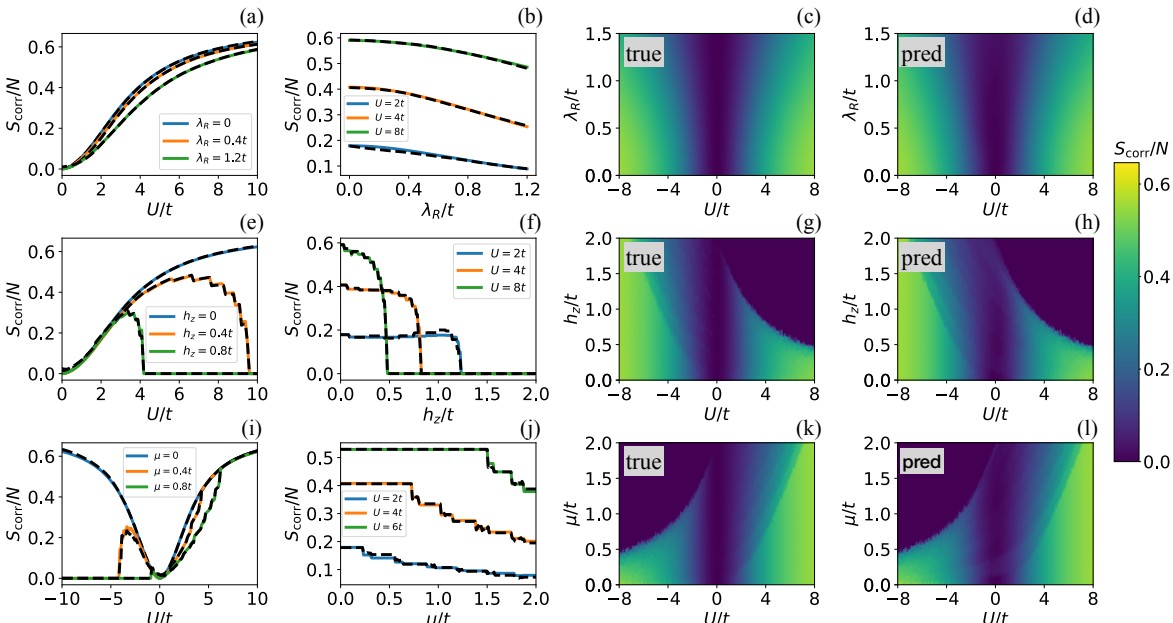

Figure 2: Correlation entropy for the generalized interacting model in the limit $V = t' = 0$. Dashed lines correspond to neural-network prediction, solid lines to the real value of the correlation entropy. Dependence of $S_{\mathrm{corr}}$ on the on-site interaction $U$ for various values of $\lambda_R$ in (a), for various values of $h_z$ in (e), and for various values of $\mu$ in (i). Dependence of $S_{\mathrm{corr}}$ on $\lambda_R$ in (b), on $h_z$ in (f), and on $\mu$ in (j) for various values of $U$. Panels (c,g,k) show the real correlation entropy and (d,h,l) the neural-network prediction based on local measurements, in the plane $\lambda_R U$ (c,d), $h_z U$ (g,h), and $\mu U$ (k,l).

the many-body state can be written as a product state $|\Psi_0\rangle = \prod_j \hat{c}_j^\dagger |0\rangle$, also known as a Slater determinant, then all the eigenvalues of the correlation matrix are either 0 or 1 and the correlation entropy vanishes. A nonzero value of the correlation entropy implies an obstruction of finding a basis where $|\Psi_0\rangle$ can be written as a single product state, thus implying finite multi-particle entanglement. In a non-interacting system, the full many-body wavefunction can always be written as a product state in the diagonal single-particle bases. It is necessary to have a finite many-body interactions to produce eigenstates with positive correlation entropy. However, stronger interactions do not necessarily imply larger correlation entropy, as interactions can also drive a system towards a symmetry broken product state. The correlation entropy is thus a measure of how far away a given state is from a single product state. Thus, it is a measure of quantum complexity rather than a measure of the strength of interactions. As a proxy to many-body correlations, the correlation entropy is qualitatively different from bipartite entanglement entropies [31, 32]. Despite the fact that correlation matrices also play a role for calculating free fermion bipartite entanglement entropies [33], the correlation entropy and partition entropies characterize fundamentally different type of quantum correlations. In particular, the partition entanglement entropies are nonzero even for free fermion systems [17–19], in contrast to the correlation entropy.

To make the previous discussion concrete, we consider a one-dimensional generalized model

of spinful interacting fermions of the form

$$
\begin{aligned}
H = &-t \sum_{j,s} \left( c_{j,s}^{\dagger} c_{j+1,s} \right) + t' \sum_{j,s} \left( c_{j,s}^{\dagger} c_{j+2,s} \right) \\
&- \mu \sum_{j} \left( n_{j,\uparrow} + n_{j,\downarrow} \right) + h_z \sum_{j,s,s'} c_{j,s}^{\dagger} \sigma_z^{s,s'} c_{j,s'} \\
&+ U \sum_{j} \left( n_{j,\uparrow} - \frac{1}{2} \right) \left( n_{j,\downarrow} - \frac{1}{2} \right) \\
&+ V \sum_{j} \left( n_{j,\uparrow} + n_{j,\downarrow} - 1 \right) \left( n_{j+1,\uparrow} + n_{j+1,\downarrow} - 1 \right) \\
&+ i \lambda_R \sum_{\langle jk \rangle ss'} c_{j,s}^{\dagger} \left( \boldsymbol{\sigma}_{ss'} \times \boldsymbol{d}_{jk} \right)_z c_{k,s'} + \text{h.c.},
\end{aligned}
\tag{3}
$$

where $n_{j,s} = c_{j,s}^{\dagger} c_{j,s}$ is the number operator with spin $s$. The parameter $t, t'$ control the first and second nearest-neighbour hopping, $\mu$ the chemical potential, $U$ the on-site Hubbard many-body interaction, $V$ the nearest-neighbour electronic many-body interaction, $h_z$ the magnetic field in the $z$ direction, $\lambda_R$ the Rashba spin-orbit coupling (SOC), $\boldsymbol{d}_{jk} = \boldsymbol{r}_j - \boldsymbol{r}_k$ with $\boldsymbol{r}_j$ the location of site $j$, $\langle jk \rangle$ denotes first neighboring sites, and $\boldsymbol{\sigma} = (\sigma_x, \sigma_y, \sigma_z)$ are the spin Pauli matrices. We find the ground state of Eq. (3) using the tensor-network matrix-product state formalism [34–37], which allows extracting the different two-point correlators in the full system and evaluate the correlation entropy. Below we consider finite systems with 24 lattice sites to generate phase diagrams and the training data for the neural network. As mentioned above, the correlation entropy is an extensive quantity. Therefore we consider the correlation entropy density, which becomes system-size independent for sufficiently large systems $L \gtrsim 20$. The employed systems with 24 sites are thus observed to accurately reproduce the phase diagram of the Hubbard model in the thermodynamic limit.

## 3 Results

### 3.1 Correlation entropy for extended Hubbard model

The correlation entropy in the $UV$-plane is presented in Fig. 1(b) and reproduces the essential features of the phase diagram as obtained from the von Neumann entanglement entropy [15, 17, 19, 20, 38–41]. For strong interactions $U, V \gg t$, the $UV$-phase show three different phases, charge-density wave, spin density wave, and a phase separation [15, 17, 19, 20, 38–41]. The observation of distinct phases in the phase diagram demonstrate that the correlation entropy allows identifying different correlated states. As expected, for vanishing interactions $U = 0$ and $V = 0$, the correlation entropy vanishes. The on-site interaction $U$ acts as the main driver of the correlation entropy which saturates to the value $\ln 2$ per site at large $|U|$ [27]. In contrast, a strong nearest neighbour coupling $V$ tends to suppress quantum correlations. The reason for this is that $V$ promotes a charge density wave state for $V > 0$ and a phase-separated state for $V < 0$, both of which are described by a product state for large $|V|$ [17, 19, 20, 42]. While both of these states are drastically different from the non-interacting ground state, they do not support sizable many-body quantum entanglement, as indicated by the vanishing correlation entropy.

## 3.2   Inferring correlation entropy from local measurements

Having established the main features of the correlation entropy on our model, we now present how to extract from a limited set of measurements. The central idea relies on reverse-engineering the correlation entropy from a small number of local correlation functions [43–49]. The many-body entanglement entropy is notoriously difficult to access in experiments [50] and measuring the correlation entropy faces similar challenges as it would require knowledge of the full correlation matrix [51]. With access to the full correlation matrix, the correlation entropy can be directly computed. However, this is experimentally unfeasible for large systems. This limitation motivates finding a complementary strategy involving correlations only between a limited number of lattice sites. In the following, we show that correlations in a few sites are sufficient to reverse-engineering the correlation entropy using a deep learning approach.

A schematic of our methodology is shown in Fig. 1(a). The input layer corresponds to a set of correlation functions of the spinful interacting fermionic chain measured locally. As inputs, we consider single-particle correlators in Eq. (1) and density-density correlators

$$f_{ij}^{ss'} = \langle \Psi_0 \, | n_{is} n_{js'} | \, \Psi_0 \rangle \tag{4}$$

evaluated on the four sites at the center of the chain. The indexes $i, j$ in the correlators run over the four neighbouring sites in the center of the chain. Our algorithm combines both types of correlators in the supervised learning algorithm, as the two types of correlators provide complementary information of the many-body state [52]. Details of the neural network architecture can be found in Methods. Crucially, we note that the neural network algorithm employing the training data from the 24-site chain already accurately reproduces the correlation entropy in the thermodynamic limit. Therefore, to apply the method to thermodynamically large systems, it is still sufficient to use input data from four adjacent sites and train the algorithm with only 24-site model. This summarizes the remarkably power of the algorithm: the size of the subsystem from which the input data is obtained and the size of the system used for the train the algorithm do not scale as the size of the physical system which can be macroscopic.

As a specific benchmark, we first consider a minimal model with only nonzero $\{t, U, V\}$ in Eq. (3), which corresponds to the usual one-dimensional Hubbard model. As shown in Fig. 1(c), the machine learning prediction reproduces the exact results with high accuracy. Similarly, we also apply this methodology to an interacting model with Hubbard interaction, SOC, magnetic field, and doping separately. The results are shown in Fig. 2. In the left two columns, the dependence of $U$ and other parameters on the correlation entropy are given with the prediction of the neural network model as the black dashed curves. We can see that the prediction of the neural network model captures well the features of the correlation entropy in the different regimes, as shown in the right-most column in Fig. 2. These results show the flexibility and accuracy of our methodology for various regimes of the interacting model.

## 3.3   Reconstructions from partial correlators and robustness to noise

In the section above we showed that the deep learning approach provides accurate results for special cases of Eq. (3) using simultaneously two-point and four-point correlators. However, experimentally, extracting one of the two sets of correlators may be more challenging than the other in specific setups. Furthermore, in experimentally realistic scenarios, the extracted correlators are expected to have a finite amount of noise. To address those challenges, here we show how this algorithm can be implemented with either two-point or four-point correlators separately, and that it is robust to noisy data.

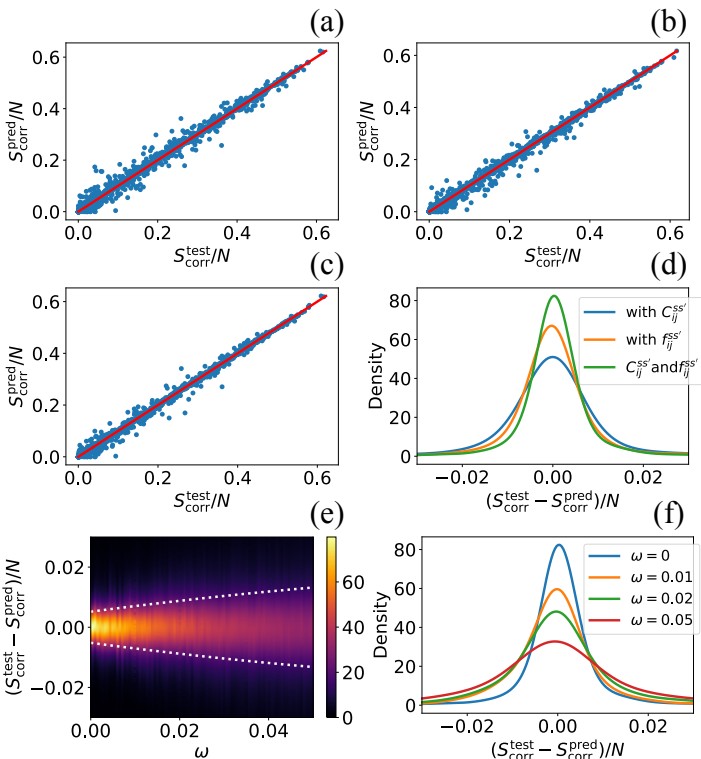

Figure 3: Predictions of the neural-network algorithm on the generalized interacting model in Eq. (3). (a) Predictions of the model trained with particle-particle correlators $C_{ij}^{ss'}$. (b) Predictions of the model trained with density-density correlators $f_{ij}^{ss'}$. (c) Predictions of the model trained on both $C_{ij}^{ss'}$ and $f_{ij}^{ss'}$. (d) Probability distribution of the difference between the predicted entropy from local measurements and the true one $S_{\text{corr}}^{\text{test}} - S_{\text{corr}}^{\text{pred}}$ in (a,b,c). (e) Probability distribution of the error in the prediction in the presence of noise. The half band width of the probability distribution is depicted as white dotted curve. (f) Probability distribution of the error in the prediction for particular noise rates.

We first consider the full model in Eq. (3) and a supervised algorithm that have been trained separately with local two-point correlators $C_{ij}^{ss'}$ or four-point density-density correlators $f_{ij}^{ss'}$. The results are shown in Fig. 3(a) and (b), highlighting a good agreement between the true end predicted correlation entropies. Both types of local correlators lead to comparable accuracy, yet lower than the model trained in both. The neural network model trained on both two-point $C_{ij}^{ss'}$ and four-point $f_{ij}^{ss'}$ correlation functions is shown in Fig. 3(c). This leads to a higher accuracy compared to the cases with only a single type of input data. More quantitative comparison, a probability distribution [1] of the error between the true and predicted results, is shown in Fig. 3(d). We observe that the most accurate results are obtained with the network trained on both $C_{ij}^{ss'}$ and $f_{ij}^{ss'}$, leading to typical error of 1%.

The robustness of the neural-network model can be benchmarked by introducing random noise in the data. For every different realization, we consider a different random noise that modifies

---

[1]The probability distribution is computed with kernel density estimation algorithm

every single correlator $\Lambda_{ij}^{ss',0} = \{C_{ij}^{ss'}, f_{ij}^{ss'}\}$ independently as

$$\Lambda_{ij}^{ss'} = \Lambda_{ij}^{ss',0} + \chi_{ij}^{ss'}, \tag{5}$$

with $\Lambda_{ij}^{ss',0}$ the original correlators, $\Lambda_{ij}^{ss'}$ the noisy correlators, $\chi_{ij}^{ss'}$ the random noise between $[-\omega, \omega]$, and $\omega$ controlling the amplitude of the noise. A heat plot of the probability distribution of the error of the prediction of the neural network model is shown in Fig. 3(e). The probability distribution for particular values of $\omega$ is shown in Fig. 3(f). We can see that the neural network model is robust for a noise rate up to $\omega = 0.02$. The accuracy is comparable with the model trained in the absence of numerical noise. For stronger noise up to $\omega = 0.05$, the neural-network model predicts the correlation entropy within the error of 1.5%.

Obtaining information on multi-particle quantum correlations and entanglement with experimental probes is, in general, prohibitively challenging due to the large number of correlators that should be measured. With the machine learning algorithm introduced above, the correlation entropy can be deduced from correlation functions on a handful of lattice sites. Our results show that the correlation entropy for macroscopically large system can be extracted using local correlators obtained in a small number of lattice sites (four sites here) as the input. In particular, for a non-interacting system the correlation entropy is identically zero, and thus our methodology allows characterizing the correlations stemming purely from many-body interactions. From the physical point of view, single-particle correlators can be directly inferred from non-local transport measurements [53, 54]. Spin-resolution in the correlators can be directly obtained by using spin-polarized leads [55, 56]. Regarding four-point correlators, our method employs density-density correlators that can be directly extracted from Friedel oscillations of the electronic charge when impurities are deposited on a single site [57, 58]. Analogously, spin-resolution could be achieved by employing magnetic impurities [59, 60].

## 4   Summary

To summarize, we proposed a supervised learning approach that allows to quantitatively extract bulk quantum correlations in interacting fermionic many-body systems from local measurements. Focusing on a generalized interacting spinful model, we demonstrated that the correlation entropy can be accurately inferred from a modest set of bulk measurements. The mapping between the local correlators and the correlation entropy is implemented with a neural-network algorithm. Furthermore, we demonstrated that this strategy can be implemented with two-point correlators, four-point-correlators, and both, providing a flexible strategy that can be adapted to different experimental setups. Finally, we demonstrated that this algorithm is robust to noise in the measurements, highlighting its feasibility for applications to experimental data. Our work provides a starting point to quantitatively characterize quantum correlation from experimental data in electronic systems, allowing us to map the degree of quantum entanglement from local measurements in quantum materials.

## 5   Methods

Here we discuss the architecture of the neural network used, that were implemented with Keras [61]. Accounting for spin, for each set of parameters $\{t, t', \mu, U, V, h_z, \lambda_R\}$, the input layer consists of 32

one particle correlators $C_{ij}^{ss'}$ and 32 density-density correlators $f_{ij}^{ss'}$. The training data is obtained by randomly generating 12,000 examples for sets of parameters $\{t, t', \mu, U, V, h_z, \lambda_R\}$ of a 24-site spinful interacting fermionic chain. After solving the ground state for each set of parameters, we can evaluate the correlators which are used as the input data, as well as the resulting correlation entropy in Eq. (2) employed as the target value for the supervised learning algorithm. We consider three hidden layers in the architecture with the sequence of nodes 1024/2048/1024, that leads to the output layer, the correlation entropy. We choose Adam algorithm for optimizer, with the learning rate $5 \times 10^{-5}$, and mean absolute error as the loss function. We target the validation loss below 0.005 in the training process which results similar accuracy on the test set. The neural-network architecture can be found in Zenodo [62].

We now elaborate on the many-body algorithms used. Many-body calculations were performed using the tensor-network matrix-product state (MPS) formalism implemented in dmrgpy [37], based on ITensor [35, 36]. Correlators are performed with full tensor contraction of the MPS [63] and ground states are computed with the density-matrix renormalization algorithm [34]. The spinful many-body fermionic problem is mapped to a spin problem using a spinful Jordan-Wigner transformation with Jordan-Wigner strings. The data generation based on tensor networks can be found in Zenodo [62].

# 6 Code availability

The data and codes implementing the data generation and training are available in Zenodo [62].

# 7 Acknowledgements

JLL acknowledges the financial support from the Academy of Finland Projects No. 331342 and No. 336243 and the Jane and Aatos Erkko Foundation. T.O. acknowledges the Academy of Finland project 331094 for support. FA acknowledges the financial support from the Magnus Ehrnrooth foundation. FA and JLL acknowledge the computational resources provided by the Aalto Science-IT project. We thank R. Koch for useful discussions.

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
