# Peer review of "Extracting electronic many-body correlations from local measurements with artificial neural networks"

_SciPost Physics_

## Round 2 · Referee Report · Anonymous (Referee 1) · 2022-12-5

Strengths

1) The article is in general very clear. 2) The method is simple and correctly reproduce the expected results. 3) Significance for the analysis of experimental data.

Weaknesses

1) Although the shortcut provided by the artificial network may be interesting, it is not "groundbreaking". 2) Lack of new physical results, and a study in 2D would be interesting. 3) The robustness to noise is not convincing. 4) It is not clear if the thermodynamic limit is reached, and how the algorithm converges with an increasing number of input correlators.

Report

The authors study the correlation entropy $S_{\mathrm{corr}}$ of one-dimensional Hubbard and extended Hubbard models, by means of artificial neural networks. They begin with a pedagogical introduction of the correlation entropy defined through eigenvalues of the one-body density matrix (called here the correlation matrix), and discuss its ability to quantify the distance of a state to a single product state, or in other words to quantify quantum correlations. They introduce the extended Hubbard model with unusual Rashba spin-orbit coupling. Then, they train an artificial neural network on a large set of ground states obtained with MPS+DMRG simulations to find $S_{\mathrm{corr}}$ for a given set of parameters. The trained network is able to reproduce the correlation entropy of a system of $L=24$ sites with the knowledge of a reduced number of two-body and density-density correlators involving only four adjacent sites. Finally, the authors consider noise in the input correlators to demonstrate the robustness of their process to small fluctuations.

The results are interesting and the method looks appealing for the analysis of some specific experimental data to extract the correlation entropy of a given system, however, I don't believe that it could have a wider impact. Hence, I think that the present article does not meet the acceptance criteria required in SciPost Physcics, but the results stay interesting and may be published in another journal. I nevertheless have some remarks for the authors:

  • The introduction is too succinct: it lacks motivations concerning the model (that is not exactly the usual one), and of an introduction to deep learning in these systems, which was already investigated in previous studies: the benefits of such a method should be clearly stated at the beginning.

  • It is stated that the correlation entropy density becomes system-size independent for systems with $L\geq 20$. However, when correlations are strong in a many-body system, the energy scales at play are small and require large systems to be reached, and the larger the better. It would be interesting to have some scaling in $L$ to see how does the correlation entropy density evolves when the system size is increased, as MPS simulations can reach several hundreds of sites in 1D. These small systems (tens of sites) could be nevertheless interesting to describe engineered optical lattices or quantum dot arrays. Note that in this sense, an extension to 2D system would add significance to this work.

  • The study of the noise is not convincing as is presented here: indeed neural networks are very sensitive to noise in the data, and Fig. 3 (f) seems to confirm that the predicted result are highly affected by the noise: it would be interesting to have the half band-width plotted against $\omega$ and push the simulations to larger noise to see when the breakdown appears. It would be also interesting to investigate the robustness to noise depending on the degree of correlations in the system, and do this curve for different values of $S_{\mathrm{corr}}$.

Regarding small details: - In Part2 : "tensor network matrix product states..." is redundant and could be replaced by "a tensor network variational state / tensor-network formalism", since tensor networks states are the generalization of MPS in higher dimensions. - In Fig 3: Figs. (d) and (f) are sufficient, Figs. (a),(b),(c) and (e) add no information and burden the text. - In Fig. 3 (e)-(f): Is $\omega$ in units of $t$ ? What parameters are used for the Hamiltonian? It is important to know the amount of correlations in the system for the study of noise. - Some literature concerning deep learning in the Hubbard model is missing: J. Phys. Soc. Jpn. 86, 093001 (2017), J. Phys. Soc. Jpn. 88, 065001 (2019) (may not be exhaustive).

Requested changes

See the different points in the report.

  • validity: high
  • significance: good
  • originality: ok
  • clarity: top
  • formatting: good
  • grammar: excellent

Author:  Faluke Aikebaier  on 2022-12-23  [id 3182]

(in reply to Report 1 on 2022-12-05)
Category:
answer to question
correction

Response to Reviewer 1

The authors study the correlation entropy S_{corr} of one-dimensional Hubbard and extended Hubbard models, by means of artificial neural networks. They begin with a pedagogical introduction of the correlation entropy defined through eigenvalues of the one-body density matrix (called here the correlation matrix), and discuss its ability to quantify the distance of a state to a single product state, or in other words to quantify quantum correlations. They introduce the extended Hubbard model with unusual Rashba spin-orbit coupling. Then, they train an artificial neural network on a large set of ground states obtained with MPS+DMRG simulations to find S_{corr} for a given set of parameters. The trained network is able to reproduce the correlation entropy of a system of L=24 sites with the knowledge of a reduced number of two-body and density-density correlators involving only four adjacent sites. Finally, the authors consider noise in the input correlators to demonstrate the robustness of their process to small fluctuations.

The results are interesting and the method looks appealing for the analysis of some specific experimental data to extract the correlation entropy of a given system, however, I don't believe that it could have a wider impact. Hence, I think that the present article does not meet the acceptance criteria required in SciPost Physics, but the results stay interesting and may be published in another journal. I nevertheless have some remarks for the authors:

Our Response

We thank the referee for the positive feedback on our work. The main message of our paper, demonstrated in detail for the generalized Hubbard model, is that the artificial neural network and the developed protocol enable one to reverse engineer the correlation entropy (which characterizes the strength of the many-particle entanglement) from a small number of correlation function measurements. A priori, to evaluate the correlation entropy, one would need correlation function data from a thermodynamically large system, which is experimentally challenging, if not unfeasible. The ANN can overcome the crucial bottleneck to extract the correlation entropy from experimental data. Our work provides a starting point for obtaining experimentally inaccessible many-body quantities with the help of ANN. We consider our work has a wider impact in such a sense, and suitable for SciPost Physics.

  • The introduction is too succinct: it lacks motivations concerning the model (that is not exactly the usual one), and of an introduction to deep learning in these systems, which was already investigated in previous studies: the benefits of such a method should be clearly stated at the beginning.

Our Response

We thank the Referee for the suggestion, we have now included further details about the model and the advantages of supervised learning procedures.

  • It is stated that the correlation entropy density becomes system-size independent for systems with L≥20. However, when correlations are strong in a many-body system, the energy scales at play are small and require large systems to be reached, and the larger the better. It would be interesting to have some scaling in L to see how does the correlation entropy density evolves when the system size is increased, as MPS simulations can reach several hundreds of sites in 1D. These small systems (tens of sites) could be nevertheless interesting to describe engineered optical lattices or quantum dot arrays. Note that in this sense, an extension to 2D system would add significance to this work.

Our Response

We thank the Referee for the suggestion. We fully agree with the Referee that the finite size effect could appear, nonetheless, we have verified that our calculations are effectively within the thermodynamic limit within our numerical accuracy as shown in Appendix 8.1. We elaborate on the details below.

In the manuscript, we targetted an algorithm trained on 24 sites with input data from 4 lattice sites in the center, arguing that accurately reproduces S_{corr}/N for thermodynamically large systems. The first evidence that this is the case stems from the fact that the predicted S_{corr}/N essentially reproduces the thermodynamic phase diagram. We note that if the physical system is very small, finite-size effects would play a role and potentially impact the prediction of the correlation entropy. Furthermore, for the sake of concreteness, we now present a systematic study of the finite-size dependence of the correlation entropy on the system size is seen in the provided figure in Fig. 4 in Appendix 8.1. In particular, it is clearly observed that for 24 sites the correlation entropy has effectively converged with system size, as expected from a thermodynamic limit. We want to note that there could be small corrections to the correlation entropy, yet as shown in our calculations, such corrections would be much smaller than the numerical inaccuracy of the extraction of the correlation entropy shown in Fig. 3. We finally would like to point out that our methodology is also applicable in cases in which finite size effects have a sizable contribution such as engineered optical lattices or quantum dot arrays as the Referee mentions. We have added a note about this in our manuscript.

Regarding the extension to 2D systems, there are no conceptual obstacles for our approach to be applied to 2D models in the future. The major challenge in this situation is to generate the training data, which requires a methodology to solve a relatively large two-dimensional interacting model. While some specific models can be solved using Quantum Monte Carlo solvers, generic models would present the sign problem, limiting the training systems that could be generated. It is worth noting that future development of neural-network quantum states may allow solving a generic set of interacting fermionic models, that would allow generating our training data. In contrast, for interacting one-dimensional systems, we could solve a generic large set of models using matrix-product states, allowing us to generate all the required training data. We have added a note about this in the manuscript. We added a related discussion in the Method section.

  • The study of the noise is not convincing as is presented here: indeed neural networks are very sensitive to noise in the data, and Fig. 3 (f) seems to confirm that the predicted result are highly affected by the noise: it would be interesting to have the half band-width plotted against ω and push the simulations to larger noise to see when the breakdown appears. It would be also interesting to investigate the robustness to noise depending on the degree of correlations in the system, and do this curve for different values of S_{corr}.

Our Response

We thank the Referee for the suggestion. In our revised manuscript, we have added a more detailed study on noise in appendix 8.2. We have increased the range of $\omega$ in the heat plot of the mean error of the prediction, including the dependence of the half bandwidth on $\omega$. We have also shown the robustness of the model to noise in terms of the degree of correlation against the mean absolute percentage error. As shown in Fig. 5.1, we observe that, although the noise slightly increases the error, the predictions of the network remain robust. In our revised manuscript, we have addressed the robustness of noise depending on the degree of correlations as suggested by the Referee. As shown in Fig. 5.2, it is observed that the relative error in the entropy prediction is the strongest for small entropies. We have added new figures and this discussion in the revised manuscript.

Regarding small details: - In Part2 : "tensor network matrix product states..." is redundant and could be replaced by "a tensor network variational state / tensor-network formalism", since tensor networks states are the generalization of MPS in higher dimensions.

Our Response

We have rephrased this point in the manuscript.

  • In Fig 3: Figs. (d) and (f) are sufficient, Figs. (a),(b),(c) and (e) add no information and burden the text.

Our Response

We thank the Referee for the note. Figure 3 (a)-(c) shows the model accuracy trained on two-point or/and four-point correlators in terms of values of S_corr. With these figures, we show that the ANN model can generate correlation entropy separately on the two kinds of correlators and works best with both kinds. Figure 3 (e) shows the robustness of the model against numerical noise, including the dependence of the robustness on ω. Figure 3 (f) shows special cases of Figure 3 (e). We believe that the more general case in Figure 3 (e) helps visualize the robustness of the neural network model, and for this reason, we consider that it will be helpful to the reader. Given all the points above, we thank the Referee for the suggestion, yet we believe that including those plots is informative and thus we kept them in the figure.

  • In Fig. 3 (e)-(f): Is ω in units of t ? What parameters are used for the Hamiltonian? It is important to know the amount of correlations in the system for the study of noise.

Our Response

1) The unit of ω is unity as it represents the strength of numerical noise. 2) The parameters are shown in the Hamiltonian in Eq.(3). The values of these parameters for obtaining the > data are generated randomly. 3) We have added a figure regarding the degree of correlation against the noise rate in Fig.5(b).

  • Some literature concerning deep learning in the Hubbard model is missing: J. Phys. Soc. Jpn. 86, 093001 (2017), J. Phys. Soc. Jpn. 88, 065001 (2019) (may not be exhaustive).

Our Response

We thank the Referee for bringing up these references, and we have included them in the revised version of our manuscript.

---

## Round 2 · Referee Report · Izak Snyman (Referee 2) · 2022-12-6

Strengths

1) The paper contains an interesting result regarding the nature of many-body correlations: in the studied system, a few one-body and density-density correlators serves as a fingerprint that can identify something about the nature of the ground state of the system, namely, how far from a single Slater determinant it is. 2) The presentation is clear and uncluttered. 3) A substantial numerical analysis was performed.

Weaknesses

1) The utility of the result for future experiments is probably overestimated.

Report

In the submitted manuscript, the authors use DMRG to solve an interacting 1D chain of 24 sites with a 6-dimensional parameter space. The central quantity of study is the Von Neumann entropy of the one-particle density matrix, which measures how far a fermionic wave function is from a Slater determinant. An exact evaluation requires knowledge of the full one-particle density matrix. However, exploiting data generated with the DMRG solver, the authors train a neural network to guess the entropy with remarkable accuracy from knowledge of a limited set of expectation values of single- and two-particle operators. The authors suggest that this paves the way for experimentalists studying correlated electronic systems to employ machine learning to determine the Von Neumann entropy of the one-particle density matrix, and thus the extent of non-trivial many-body correlations, from a limited number of measurements.

The authors’ success at reconstructing the Von Neumann entropy of the one-particle density matrix across the whole phase diagram, from limited data, is an interesting result, that cost a substantial numerical effort. It could lead to advances in our understanding of the nature of quantum correlations in interacting many-body systems. In my opinion it may warrant publication in SciPost. I do however think that the current version leaves several important questions unanswered, and that this needs to be addressed, before a final decision can be made regarding whether the manuscript is accepted or rejected.

Requested changes

Please answer the following questions, and where appropriate, update the manuscript to address the issue:

1) How can one train a neural network to predict the correlation entropy if one does not have a quantitatively accurate solver for the microscopic model of the system which one is studying experimentally? If this is a stumbling block, then the authors’ idea will only work on highly idealised one-dimensional models, that cannot be realized experimentally with sufficient fidelity for the scheme to work, and which are anyway understood well enough “digitally” that it is not necessary to perform an “analogue” determination of the correlation entropy.

2) How would one in principle measure the off-diagonal one-particle correlator <c_is^\dagger c_js’> ? In a large, clean system, the one-particle density matrix is diagonal in crystal momentum p. Is it more difficult to measure <c_ps^\dagger c_ps’> than <c_is^\dagger c_js’>? If not, what would prevent one from measuring the latter at a high enough p-resolution to calculate the correlation entropy directly?

3) How complicated is the position-representation one particle density matrix <c_is^\dagger c_js’> of the system that the authors study? In a large system, it depends only on the distance i-j, and the spin indices. I could imagine that the dependence in each phase, is either a power law or exponential decay with respectively a characteristic exponent or decay length that could indeed be extracted from a few measurements, after which the whole matrix can be extrapolated. Is that how the neural network essentially operates?

4) I also have the following technical question. In the methods section, it says that 32 one-particle correlators and 32 density-density correlators were used. This number is probably obtained as one half of the entries of an 8 by 8 matrix (four positions and two spin values). My first question is, should this not be 36 (the independent entries for an 8x8 hermitian matrix?) Secondly, if the authors are indeed close to the thermodynamic limit, I would expect only the distance i-j to matter in the correlators. This would allow one to get the same result from only 11 one-particle correlators and 9 density-density correlators. Can the authors please comment on why they seem to include several (i,j) measurements with the same i-j? I am worried that the one-particle density matrix is further from the thermodynamic limit than is claimed.

  • validity: -
  • significance: -
  • originality: high
  • clarity: high
  • formatting: excellent
  • grammar: excellent

Author:  Faluke Aikebaier  on 2022-12-23  [id 3183]

(in reply to Report 2 by Izak Snyman on 2022-12-06)
Category:
answer to question
correction

Response to Reviewer 2

In the submitted manuscript, the authors use DMRG to solve an interacting 1D chain of 24 sites with a 6-dimensional parameter space. The central quantity of study is the Von Neumann entropy of the one-particle density matrix, which measures how far a fermionic wave function is from a Slater determinant. An exact evaluation requires knowledge of the full one-particle density matrix. However, exploiting data generated with the DMRG solver, the authors train a neural network to guess the entropy with remarkable accuracy from knowledge of a limited set of expectation values of single- and two-particle operators. The authors suggest that this paves the way for experimentalists studying correlated electronic systems to employ machine learning to determine the Von Neumann entropy of the one-particle density matrix, and thus the extent of non-trivial many-body correlations, from a limited number of measurements.

The authors’ success at reconstructing the Von Neumann entropy of the one-particle density matrix across the whole phase diagram, from limited data, is an interesting result, that cost a substantial numerical effort. It could lead to advances in our understanding of the nature of quantum correlations in interacting with many-body systems. In my opinion it may warrant publication in SciPost. I do however think that the current version leaves several important questions unanswered, and that this needs to be addressed, before a final decision can be made regarding whether the manuscript is accepted or rejected.

Please answer the following questions, and where appropriate, update the manuscript to address the issue:

1) How can one train a neural network to predict the correlation entropy if one does not have a quantitatively accurate solver for the microscopic model of the system that one is studying experimentally? If this is a stumbling block, then the authors’ idea will only work on highly idealized one-dimensional models, that cannot be realized experimentally with sufficient fidelity for the scheme to work, and which are anyway understood well enough “digitally” that it is not necessary to perform an “analog” determination of the correlation entropy.

Our Response

We thank the Referee for bringing this up. Solving generic interacting one-dimensional models is doable using the matrix-product state formalism as done in our manuscript. Therefore, this is would not be a challenge from a practical point of view. We elaborate on this below.

From the experimental point of view, the only required task would be to measure the correlators. The extraction of the correlation entropy is done by training the algorithm in a large enough set of interacting many-body models that are solved exactly using tensor-network methods as shown in our manuscript. The von Neumann entropy of the correlation matrix represents the measure of correlation in terms of the eigenvalues of the correlation matrix. However, the information on the correlation is already stored in the components of the correlation matrix, i.e., the correlators. The trained neural network can make use of such information to map the correlation entropy without accessing the eigenvalues of the correlation matrix. Thus, one is able to predict the correlation entropy with a handful number of correlators. If the required number of correlators is measured, then one can extract the correlation entropy by means of the proposed neural network model. We have now emphasized this point in the manuscript.

2) How would one in principle measure the off-diagonal one-particle correlator <c_is^\dagger c_js’> ? In a large, clean system, the one-particle density matrix is diagonal in crystal momentum p. Is it more difficult to measure <c_ps^\dagger c_ps’> than <c_is^\dagger c_js’>? If not, what would prevent one from measuring the latter at a high enough p-resolution to calculate the correlation entropy directly?

Our Response

The particle correlators in our manuscript can be measured with a real-space measurement technique, and in particular scanning tunneling microscopy. The non-local correlators can be measured by depositing a local impurity in site i, and measuring the changes in site j. Manipulation of local impurities can be done with scanning tunneling microscopy, the same technique used for the measurement. Spin-dependent correlators could be measured by depositing magnetic impurities in site i, and measuring changes in site j with spin-polarized scanning tunneling microscopy. As a reference, the non-local changes in charge correlators using the previous approach have been demonstrated in (Nature Physics 4, 454–458 (2008)), and non-local changes in spin response have been demonstrated in (Nature 403, 512–515 (2000)) This procedure to measure the non-local correlator stems directly from the formulation of correlators from a perturbative point of view in terms of Green's functions (Phys. Rev. B 97, 155401 (2018)). In particular, this phenomenon is what allows us to measure Fermi surfaces using quasiparticle interference for impurities in metals. We thank the Referee for motivating this discussion, which we have included in our manuscript.

3) How complicated is the position-representation one particle density matrix <c_is^\dagger c_js’> of the system that the authors study? In a large system, it depends only on the distance i-j, and the spin indices. I could imagine that the dependence in each phase, is either a power law or exponential decay with respectively a characteristic exponent or decay length that could indeed be extracted from a few measurements, after which the whole matrix can be extrapolated. Is that how the neural network essentially operates?

Our Response

We thank the Referee for pointing this out. The mechanism noted by the Referee can be indeed one of the ways in which the neural network may be extracting the correlation entropy. We note that the actual dependence of correlators on the phases could be more complicated than those two functional forms, as a strong oscillatory behavior at short distances may also appear. The neural-network algorithm indeed captures such dependencies and maps to the correlation entropy. While in principle it would be formally possible to do such extrapolations explicitly, it is worth noting that we have verified those extrapolations are extremely sensitive to noise. In particular, we attempted to implement our algorithm using a random forest methodology and found that the algorithm was dramatically unstable and unsuitable for the prediction of entanglement entropy even with very low levels of noise. In stark contrast, the neural-network approach, as shown in our manuscript, is resilient to noise, making this approach realistic with experimental data.

4) I also have the following technical question. In the methods section, it says that 32 one-particle correlators and 32 density-density correlators were used. This number is probably obtained as one-half of the entries of an 8 by 8 matrix (four positions and two spin values). My first question is, should this not be 36 (the independent entries for an 8x8 hermitian matrix?) Secondly, if the authors are indeed close to the thermodynamic limit, I would expect only the distance i-j to matter in the correlators. This would allow one to get the same result from only 11 one-particle correlators and 9 density-density correlators. Can the authors please comment on why they seem to include several (i,j) measurements with the same i-j? I am worried that the one-particle density matrix is further from the thermodynamic limit than is claimed.

Our Response

We thank the Referee for pointing this out. Indeed, in the presence of translational symmetry, some of those correlators would be related by symmetry. It is however worth noting that the model we are considering can show broken symmetry phases and in particular charge density waves. In the presence of a charge density wave symmetry broken state, the translational symmetry of the ground state would be modified, and in particular, the correlators, in general, do not depend anymore solely on |i-j|. As a result, if the algorithm was solely trained on correlators |i-j|, this approach would seriously ignore the lack of translational symmetry of several symmetry-broken states, including charge density waves, spin density waves, and potential bond-ordered states. In order to have an algorithm capable of accounting for a model displaying those states, such as the one we consider, we decided therefore to train it with all the correlators. As a result, our methodology works for systems having the original translational symmetry, and also for ground states with symmetry-broken states. We thank the Referee for motivating this discussion, which we have included in our manuscript.

---

## Editorial Decision

resubmitted